



# S2P3-R v2.0: computationally efficient modelling of shelf seas
# on regional to global scales
Paul R. Halloran[1], Jennifer K. McWhorter[1,2], Beatriz Arellano Nava[1], Robert Marsh[3], William
Skirving[4,5]
[1]College of Life and Environmental Sciences, University of Exeter, Exeter, UK
[2]School of Biological Sciences, The University of Queensland, Brisbane, Queensland, Australia
[3]University of Southampton, National Oceanography Centre, Southampton, UK
[4]Coral Reef Watch, National Oceanic and Atmospheric Administration, College Park, MD, USA
[5]ReefSense Pty Ltd, Cranbrook, Queensland, Australia
*Correspondence to*: Paul R. Halloran (p.halloran@exeter.ac.uk)
**Abstract.** The marine impacts of climate change on our societies will be largely felt through coastal waters and
shelf seas. These impacts involve sectors as diverse as tourism, fisheries and energy production. Projections of
future marine climate change come from global models. Modelling at the global scale is required to capture the
feedbacks and large-scale transport of physical properties such as heat, which occur within the climate system,
but global models currently cannot provide detail in the shelf-seas. Version 2 of the regional implementation of
the Shelf Sea Physics and Primary Production (S2P3-R v2.0) model bridges the gap between global projections
and local shelf-sea impacts. S2P3-R v2.0 is a highly simplified coastal shelf model, computationally efficient
enough to be run across the shelf seas of the whole globe. Despite the simplified nature of the model, it can display
regional skill comparable to state-of-the-art models, and at the scale of the global (excluding high-latitudes) shelf-
seas can explain >50% of the interannual SST variability in ~60% of grid cells, and >80% of interannual
variability in ~20% of grid cells. The model can be run at any resolution for which the input data can be supplied,
without expert technical knowledge, and using a modest off-the-shelf computer. The accessibility of S2P3-R v2.0
places it within reach of an array of coastal managers and policy makers. S2P3-R v2.0 is set up to be driven
directly with output from reanalysis products or daily atmospheric output from climate models such as those
which contribute to the 6th phase of the Climate Model Intercomparison Project, making it a valuable tool for
semi-dynamical downscaling of climate projections. The updates introduced into version 2.0 of this model are
primarily focused around the ability to geographical relocate the model, model usability and speed, but also
scientific improvements. The value of this model comes from its computational efficiency, which necessitates
simplicity. This simplicity leads to several limitations, which are discussed in the context of evaluation at regional
and global scales.

## 1. Introduction

The world's coastal oceans are under increasing pressure from human activity (Doney, 2010). These shallow,
relatively accessible waters are where humans interact most with the ocean, and where marine biological activity
and diversity are often at their most intense (Mora *et al.*, 2013; Bowen *et al.*, 2016). Global Circulation and Earth
System Model projections contain neither the spatial resolution nor processes required to simulate shelf seas (Holt
*et al.*, 2009). These models have been found to contain little to no skill at simulating patterns of surface



temperature warming at spatial scales lower than 1000km (Kwiatkowski *et al.*, 2014). While at regional scales
shelf-sea models are providing extremely valuable information over short time horizons (e.g. Steven *et al.*, 2019),
the state-of-the-art in shelf-sea climate projections is either to downscale global models over small regions using
complex 3D shelf sea models (e.g. Tinker and Howes, 2020) at considerable computational expense, or downscale
large-scale projections statistically (e.g. Donner *et al.*, 2005; Van Hooidonk *et al.*, 2016). S2P3-R v2.0 aims to
bridge the gap between high-complexity small scale projections, and large-scale statistical projections which
ignore local processes and dynamics.
The underlying physical-biological model used in S2P3-R is the Shelf Sea Physics and Primary Production (S2P3)
model (Simpson and Sharples, 2012). S2P3 makes the common assumption that in many regions variability on
the shelf is dominated by atmospheric and tidal processes rather than by communication with the open ocean (e.g.
Song *et al.*, 2011; van der Molen, Ruardij and Greenwood, 2017), and consequently, represents the ocean at a
location as a 1D column of water. The physical and biological components of S2P3 are discussed below, but are
described in further detail in Simpson and Sharples (2012), Sharples et al. (2006), Sharples (2008) and summarised
in Marsh et al., (2015). S2P3-R v1.0 (Marsh, Hickman and Sharples, 2015) placed S2P3 into a spatial framework
by representing the shelf sea as a 2D array of neighbouring independent 1D columns of water. S2P3-R v2.0
addresses several the limitations in S2P3-R v1.0, which prevented it from being used effectively to downscale
large-scale reanalyses or climate projections.
**2. Overview of the underlying 1D model, S2P3**
S2P3-R v2.0 is the 2$^{nd}$ generation of regional-model development building on the 1D shelf sea model S2P3
(Sharples *et al.*, 2006). The physical component of S2P3 simulates vertical profiles of temperature, turbulence
and currents in response to tidal and wind driven mixing. The model calculates the tidal slope from the prescribed
M2, S2, N2, O1 and K1 tidal ellipses, and from this, the water's velocity (Sharples *et al.*, 2006). The stress applied
by the tides is then calculated as a function of the velocity at 1m above the seabed, the density of the seawater and
a prescribed bottom drag coefficient (Sharples *et al.*, 2006). The surface stress exerted by the wind is calculated
as a function of windspeed and direction (with respect to tides), air pressure and a windspeed-dependent surface
drag coefficient (Smith and Banke, 1975). A turbulence closure scheme calculates profiles of vertical eddy
viscosity and diffusivity as a function of current shear and vertical density (Canuto *et al.*, 2001). The surface and
bottom stress are propagated through the water column as a function of the vertical eddy viscosity, which is
derived from the turbulence closure scheme (Sharples *et al.*, 2006). S2P3 considers only the role of temperature,
not salinity, on density (Sharples *et al.*, 2006), limiting its application in cold water (where density variations are
dominated by salinity), or variable salinity settings such as near river outflows.

The biological model in S2P3 takes a lightweight and pragmatic view of representing primary production.
Phytoplankton concentrations are modelled as a function of their initial concentration, vertical mixing, growth
rate and a fixed grazing rate (Sharples, 2008). Phytoplankton growth rate is a function of the maximum growth
rate for a given temperature and nutrient availability, modified by available photosynthetically active radiation





(PAR) and maximum light utilisation rate, minus respiration at a constant rate (Sharples, 2008). Surface PAR is
set to 45% of the net downwelling surface shortwave radiation, and this decays as a function of phytoplankton
concentration and an attenuation coefficient which is dependent on whether the water column is mixed or stratified
(Sharples, 2008). Nutrient availability is a function of vertical mixing, uptake by phytoplankton and loss through
grazing, and is restored towards a constant concentration in the lowest model level (Sharples, 2008). The simple
assumptions made within the biological model align with the desire to keep the computational cost of the model
low, but also to avoid including poorly constrained processes within the model (Sharples, 2008). These
simplifications and their impacts are discussed further in Sharples (2008). In its original form S2P3 was driven
by sinusoidal timeseries of surface air temperature and pressure, relative humidity, total cloud cover and u and v
surface winds.
**3. Scientific advances from S2P3**
Version 1 of S2P3-R modified the S2P3 code and provided bash scripts to run S2P3 as a 2D array of 1D column
models to provide a computationally efficient way to simulate shelf sea physical and biological conditions (Marsh,
Hickman and Sharples, 2015). Application of this version of the model demonstrated that this simple approach to
shelf-sea modelling produced sensible patterns of temperature, stratification and primary production on the North
West European Shelf and East China and Yellow seas, and showed that the model reproduced observed year to
year variability at two sites in the English Channel (Marsh, Hickman and Sharples, 2015). The success of S2P3-
R at reproducing physical and biological structures over the recent past has motivated the developments and
evaluation presented here. The developments described here are aimed at running the model at larger spatial scales
and over longer time periods, including into the future to downscale and explore the coastal implications of future
climate change. These developments presented several practical challenges, which are discussed below.
S2P3-R v1.0 introduced spatial information into its simulation by considering local bathymetry and tidal mixing,
as well as a latitudinal dependence of the clear-sky radiation and Coriolis parameter used within the model (Marsh,
Hickman and Sharples, 2015). Application of the model over larger spatial domains was limited scientifically
because it used common timeseries of surface air temperature and pressure, relative humidity, cloud fraction and
wind velocities to drive all water columns within a simulation. S2P3-R v2.0 addresses this limitation by utilising
meteorological timeseries specific to each grid location which are generated from reanalysis or climate models
using the provided scripts (see below and the Code Availability section).

Previous iterations of the model have represented downwelling shortwave irradiance as a function of time of year,
latitude and total cloud fraction. While this approach has been applied successfully when considering the North
West European Shelf (Sharples *et al.*, 2006; Sharples, 2008; Marsh, Hickman and Sharples, 2015), total cloud
fraction cannot account for the impacts on radiation of moving between regions of different cloud type or changes
in cloud microphysics. Over climate timescales, changes in aerosol emissions, meteorology and atmospheric
chemistry will have considerable impacts on the shortwave radiation received at the sea surface (Haywood and
Boucher, 2000), which may dominate greenhouse gas driven climate signals at regional scales (Booth *et al.*, 2012).





S2P3-R v2.0 moves to prescribing the net downwards surface radiation explicitly from the reanalysis product or
climate model output from which it is driven.
Analogous to the treatment of shortwave radiation within S2P3, the net loss of heat from the surface of the ocean
in the form of longwave radiation was calculated in S2P3-R v1.0 from the temperature-dependent longwave
emission derived from the Stefan–Boltzmann equation, moderated by cloud-fraction and humidity. This approach
cannot account for spatial/temporal changes in cloud-top height and optical thickness, which have been shown to
be as important as cloud fraction in determining the radiation field (Chen, Rossow and Zhang, 2000). These factors
are of 1st order importance when relocating the model from high to low latitudes, performing simulations spanning
these latitudes, or when considering the impacts of anthropogenic aerosols and cloud-feedbacks in response to
climate change. A further limitation of inferring the downwelling longwave radiation as a function of cloud
fraction when performing long historical simulations or simulations driven from future climate projections, is that
the change in the radiation budget associated with changing greenhouse gas concentrations is not directly
accounted for. S2P3-R v2.0 revises the surface heat-loss through longwave radiation ($Q_{LongwaveNet}$) to:

$$Q_{LongwaveNet} = \varepsilon_{longwave}\sigma T^4 - Q_{LongwaveDownwards}S \qquad (1)$$

Where $\varepsilon_{longwave}$ is the long-wave emissivity (0.985), $\sigma$ is the Stefan–Boltzmann constant ($\sigma = 5.67 \times 10^{-8}$ W
m$^{-2}$K$^{-4}$), T is the temperature of the surface layer, $Q_{LongwaveDownwards}$ is the prescribed downwelling longwave
radiation at the surface, and S is a constant to account for the fact that the model is not simulating the ocean skin,
where a proportion of the longwave radiation will be absorbed and re-emitted without interacting with the water
at the depths represented by the top layer of the model.
To facilitate longer timesteps in deeper waters, S2P3-R v1.0 scaled the vertical resolution in each water column
with the water-depth. This has been revised to a fixed 2m vertical resolution in S2P3-R v2.0 to prevent variability
in level thickness introducing spatial artefacts to simulated surface water conditions. Phytoplankton growth in the
model, and therefore primary production relies on a flux of nitrate into the lowest vertical level of the model. In
S2P3-R v2.0 we move from representing this as a single value in space and time, to a value specific to each grid
box, read in from an ancillary file. A script is provided to generate this ancillary file from World Ocean Atlas
(Levitus, 1982) data (see Code Availability section).
A schematic overview of S2P3-R v2.0 is presented in Figure 1.



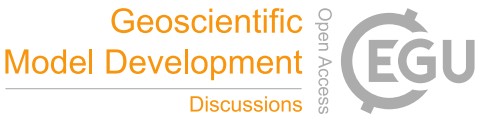

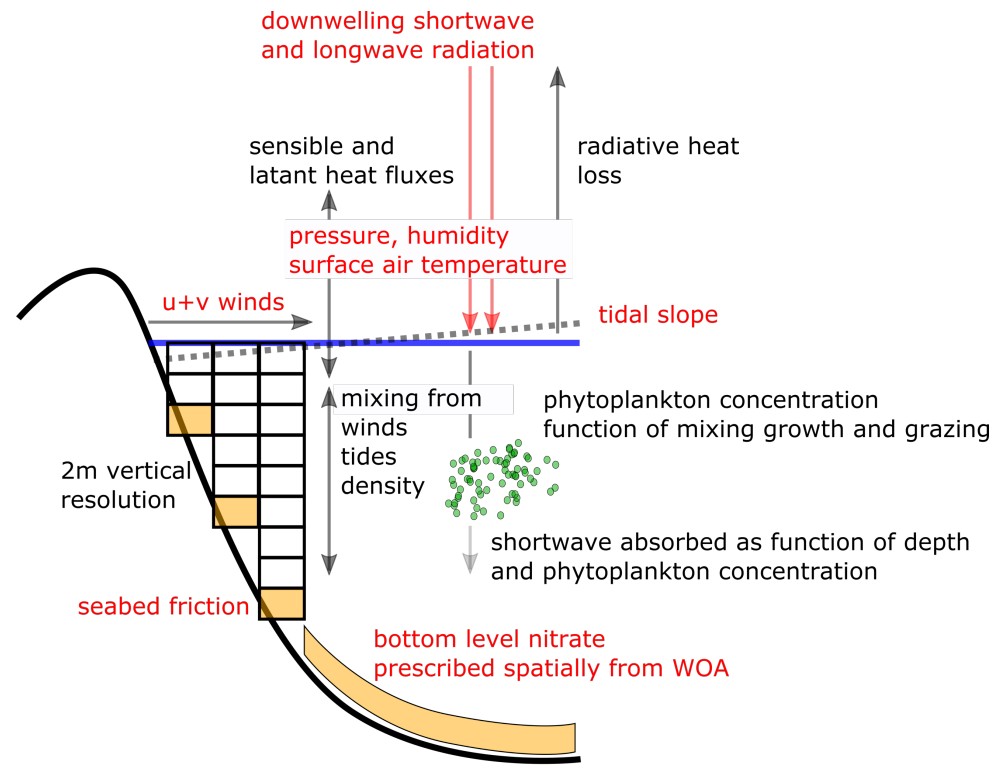

red = prescribed quantity
black = modelled quantity/process

**Figure 1. Schematic description of the processes accounted for in S2P3-R v2.0 and prescribed quantities, both forcings and constants. WOA stands for World Ocean Atlas.**

**4. Practical advances from S2P3**

The practical developments made to version 2.0 of S2P3-R fall into two categories (1) how the model runs, and (2) how to generate the data used to set up and force the model.

The initial spatial implementation of S2P3 (S2P3-R v1.0) focused on what could be achieved by running S2P3 in a regional sense, and as such provided Bash scripts which ran individual instances of the 1D model for each of the latitude/longitude locations specified in a domain file containing depth and tidal forcing data. S2P3-R v2.0 makes several changes to reduce the amount of input-output associated with this approach and distributes the processing of water columns over multiple processor cores. This is done by (1) re-writing the code which runs the underlying Fortran model code from Bash to Python using the multiprocessing module, (2) reading the depth and tidal data from file once, then passing it from memory to the Fortran code for each point, and (3) accumulating the output annually and writing this year by year to netCDF or text files. The model has been modified to run one year at a time, writing output then 'resubmitting' to allow long, high-resolution, or large spatial domain, simulations to be performed without hitting memory or submission length limits.



The independence of each grid point, combined with the developments to consolidate reading or writing data to
disk, means that the model scales very efficiently when more/fewer processor cores are used (Figure 2).

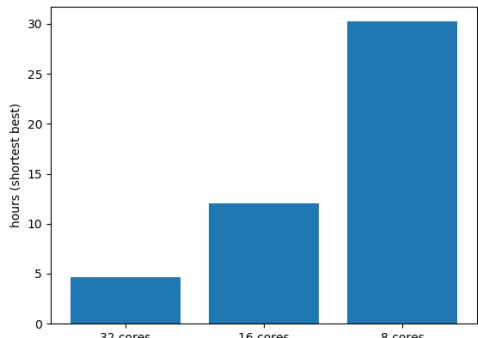

**Figure 2. Processing time in hours to complete one year of simulation at 0.2º resolution in a 'global' (65ºS-65ºN, 180ºW-**
**180ºE) configuration spanning water depths of 10-100m. The high latitudes were removed because the model assumes**
**constant salinity and the model does not include a representation of sea ice. Simulations were undertaken on an AMD**
**2990WX 32-Core 3Ghz Processor with multi-threading.**
Model developments around usability include (1) translating the Fortran code so it can be compiled with the open-
source GFortran compiler rather than the proprietary ifort compiler and by doing so improving accessibility, (2)
providing the user-option to generate output files directly in netCDF format, (3) providing an interface for
prescribing which output diagnostics the user wishes to produce, and (4) the provision of scripts and associated
ReadMe files to enable simple generation of all of the required input files (see Code Availability section). These
files are the domain (which specify the depth and tidal forcing for each model grid-point), nutrient ancillary and
meteorological forcing files (Figure 3). The input generation scripts, the input data they require, and how the
outputs are used by the main model are detailed in Figure 3. The practicalities of how to obtain and run the scripts
and associated data are detailed in the code availability section and supplied ReadMe files (see Code Availability
section).





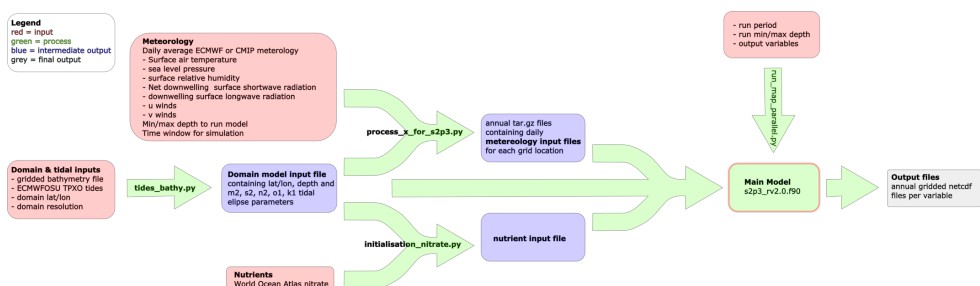

**Figure 3. Overview of the S2P3-R v2.0 framework, which includes the model and runscript but also separate scripts to**
**generate the required input files. The arrows show where externally available data or the output from one component**
**of S2P3-R V2.0 is supplied to another component/output.**
**5. Global Evaluation**
S2P3-R v2.0 is an intentionally simple model. By ignoring lateral advection, one should expect to see model
temperature biases in regions of heat convergence or divergence, i.e. where significant amounts of heat are
imported or exported through advection, or local dissipation rates are enhanced through horizonal processes. The
fact that a region may experience a temperature bias does not itself mean the model is not useful in that region.
Despite biases in average temperatures, the model may still capture variability on the timescales of interest. The
model variability may however be compromised if there is a temperature bias at low ambient temperatures, where
the non-linearity of the equation of state of seawater reduces the sensitivity of density to temperature variability.
This limits the applicability of S2P3-R v2.0 in cold waters, and alongside the specification of constant salinity
and omission of sea ice processes, means that the evaluation of the model has been restricted to the subpolar and
lower latitude ocean (<65ºN/S). The evaluation presented here is intended to allow potential model users to
identify whether S2P3-R v2.0 is an appropriate tool to use for the question and location they are interested in. We
first evaluate the global performance of the model, then focus evaluation on a mid-latitude, then a low-latitude
region. Evaluation in each section begins with the physical variables, then moves on to the biological component
of the model.
The model simulations presented here have been set up at 0.2º spatial resolution using the input fields described
in table 1.
**Table 1. Model inputs.**

| Model input | Source | Reference |
|---|---|---|
| Bathymetry Global and N.W. European Shelf | ETOPO1 | (Amante and Eakins, 2009) |
| Bathymetry Australia | 3DGBR | (Beaman, 2010) |





| Tides | Produced using the Oregon State University Tidal Inversion Software (OTPS) | (Egbert and Erofeeva, 2002) |
|---|---|---|
| Meteorological Forcing | ECMWF ERA5 | (Hersbach *et al.*, 2019) |
| Nutrients | World Ocean Atlas 13 | (Levitus, 1982) |

**5.1 Global Physical Evaluation**
An initial comparison of model SSTs against satellite SSTs (Merchant *et al.*, 2019) at a global scale indicates that
the model displays its smallest biases in the subtropics to subpolar regions (Figure 4). The prevalence of warm
biases in the tropics and cool biases in the high latitudes is consistent with export and import or warm waters from
and to these regions respectively (Figure 4). To allow potential users to examine model performance in their
regions of interest in greater detail, the data underlying Figure 4 is made available as described in the Data
Availability section.





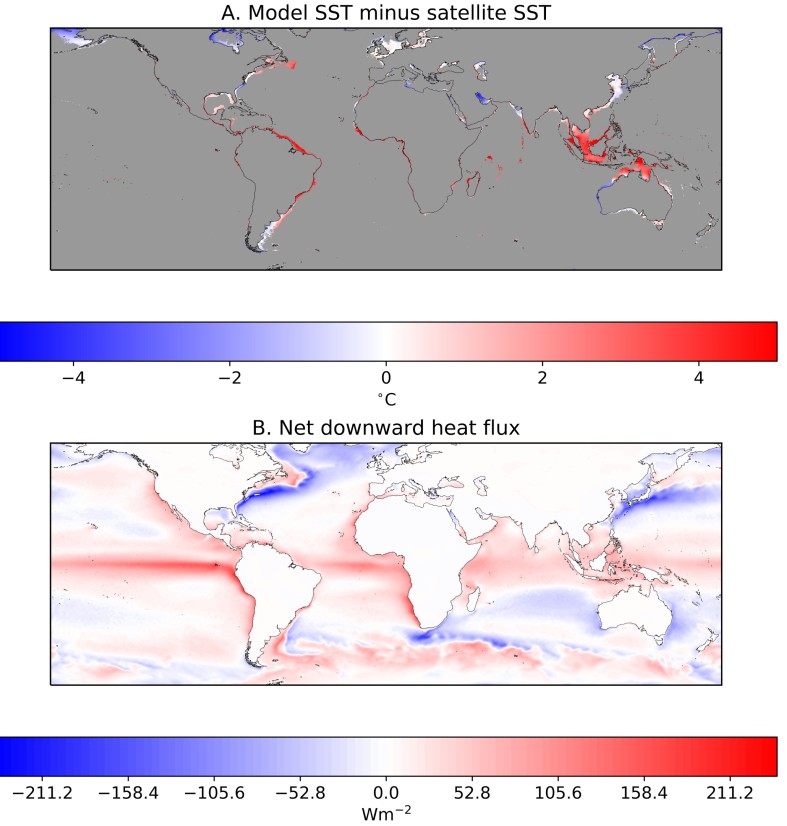

**Figure 4. A. Model SST simulation minus satellite SST data averaged between 1ˢᵗ Jan 2006 and 31ˢᵗ Dec 2016. White indicates that the model is displaying no surface temperature bias, red indicates the model displays a warm bias, and blue the model displays a cool bias. The model was forced with atmospheric data from ERA5 (Hersbach et al., 2019). B. Net surface downward heat flux calculated from the ECMWF ERA5 reanalysis (Hersbach et al., 2019). Where this is positive there is a net heat flux into the ocean, and therefore assuming the system is approximately at steady state, advection of heat out of this area. Where the net downward heat flux is negative there is advection of heat into this region. S2P3-R V2.0 does not account for lateral advection, so one would anticipate that the model will display a warm bias in regions where heat is typically advected from (i.e. tropics) and cool biases where heat is advected to (i.e. high latitudes).**

Beyond calculating the surface heat budget based on atmospheric forcings, the model skill in simulating surface temperatures comes from vertical mixing processes which exchange heat between the surface and subsurface layers as a function of temperature induced density differences and wind and tide stress. In line with this, we find that large SST biases are more prevalent at low tidal amplitudes (Figure 5). While this analysis indicates that strong tidal mixing can contribute to a skilful simulation, it does not appear that tidal magnitude provides a rule to determine where best to use this model. Stratification is highly seasonal in the mid-latitudes, with stratification typically corresponding to weak tidal mixing in the summer, and a pervasive loss of stratification during the



winter. If strong tides played a 1$^{st}$ order role in model skill, one would expect to see smaller model biases in the
summer than winter across the mid-latitudes (Figure 6).
**Figure 5. 2D histogram demonstrating the relationship between tidal amplitude (M2 tide) and absolute annual-mean**
**SST difference between the model and satellite data.**

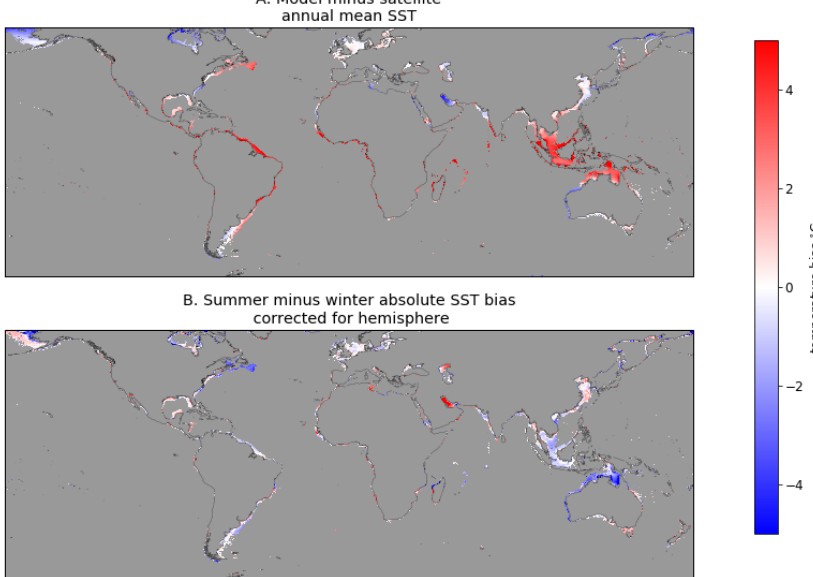

**Figure 6. Annual mean SST bias (top), and difference in absolute SST bias between summer and winter (bottom). In**
**(b) Blue indicates that the summer months (June, July, August in the Northern hemisphere and December, January,**
**February in the Southern hemisphere) display a smaller absolute bias than the winter months (December, January,**
**February in the Northern hemisphere and June, July, August in the Southern hemisphere).**





Despite the model displaying average temperature biases across some regions of up to ~3K, there is no consistent
relationship between such biases and the model's ability to correctly simulate year-to-year variability (Figure 7).
More than half of the year-to-year variability is captured by ~60% of the simulated grid cells (Figure 8). Squared
Pearson's product moment correlations ($R^2$) calculated between (i) annual mean SST timeseries at each grid point
from the ERA5 forced S2P3R V2.0 simulations and (ii) satellite SST data (Merchant *et al.*, 2019) from 2006-
2016 (inclusive) demonstrate high levels of skill in areas such as north of Australia, the Java Sea and the Bering
Sea (Figure 7) despite these areas displaying significant positive/negative temperature biases (Figure 4).
Conversely, the northern South China sea and southern Australia display low skill at capturing interannual
variability (Figure 7), despite the model displaying low temperature biases in these regions (Figure 4). In the case
of the South China sea this may relate to highly variable riverine freshwater influences on stratification.

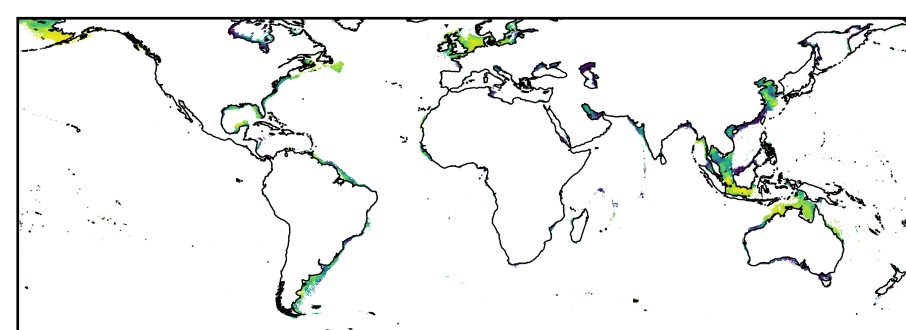

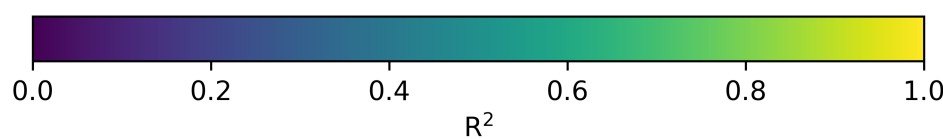

**Figure 7. Pearson's $R^2$ calculated between annual mean Model SST simulation and annual mean satellite SST data**
**(Merchant et al., 2019) between 2006 and 2016.**



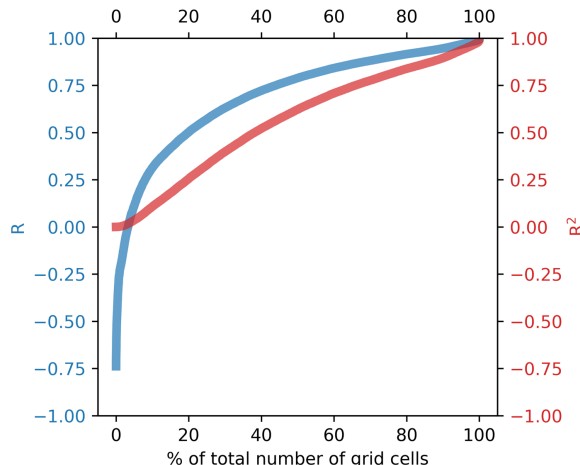

**Figure 8. Sorted R and R² values from all grid cells calculated from global shelf-sea SST simulation correlation with**
**satellite SST (Figure 7).**
**5.2 Global Biogeochemical Evaluation**
The biological component of S2P3 v2.0 remains unchanged from previous versions, other than through the
addition of a spatially varying nutrient field derived from World Ocean Atlas (Levitus, 1982) to which the bottom
water nitrate is relaxed. S2P3 has previously been used to investigate biological questions including investigating
the drivers of timing of spring blooms in response to stratification (Sharples *et al.*, 2006) and to explore the impact
of tidal cycles on productivity (Sharples, 2008) for typical North West European shelf seas. More recently a
version of S2P3 has been developed to better represent the impacts of grazing and to include the impact of photo-
acclimation on phytoplankton growth (Bahamondes Dominguez *et al.*, 2020).
Evaluation of the model's biological performance at a global scale is more challenging than the evaluation of
surface temperature, because satellite chlorophyll-a products are often unreliable in shallow waters, where
suspended sediment, coloured dissolved organic matter (CDOM) and bottom reflection influence the retrievals
considerably (Darecki and Stramski, 2004). The analysis presented here uses the ESA CCI Chlorophyll-a product
data (Sathyendranath *et al.*, 2020) but filters out waters shallower than 70m (Jackson et al., 2019) to avoid the
issues mentioned above.  The model demonstrates low (<0.2 mg m⁻³) chlorophyll-a biases when compared to
satellite estimates in all regions other Southeast Asia, Australia, the Baltic Sea and the northern Bering Sea (Figure
9). The most extensive area of bias being Southeast Asia and Australia. This is also an area of high SST bias
(Figure 4), although there is no strightforward relationship between regoinas of SST and regions of chlorophyll-
a bias. Phytoplankton growth in the model is a function of, amongst other factors, temperature and
Photosynthetically Active Radiation (PAR). Overestimation of chlorophyll-a may therefore be a response to
postive seawater temperature biases, or both may be responding to a positive shortwave radiation bias.



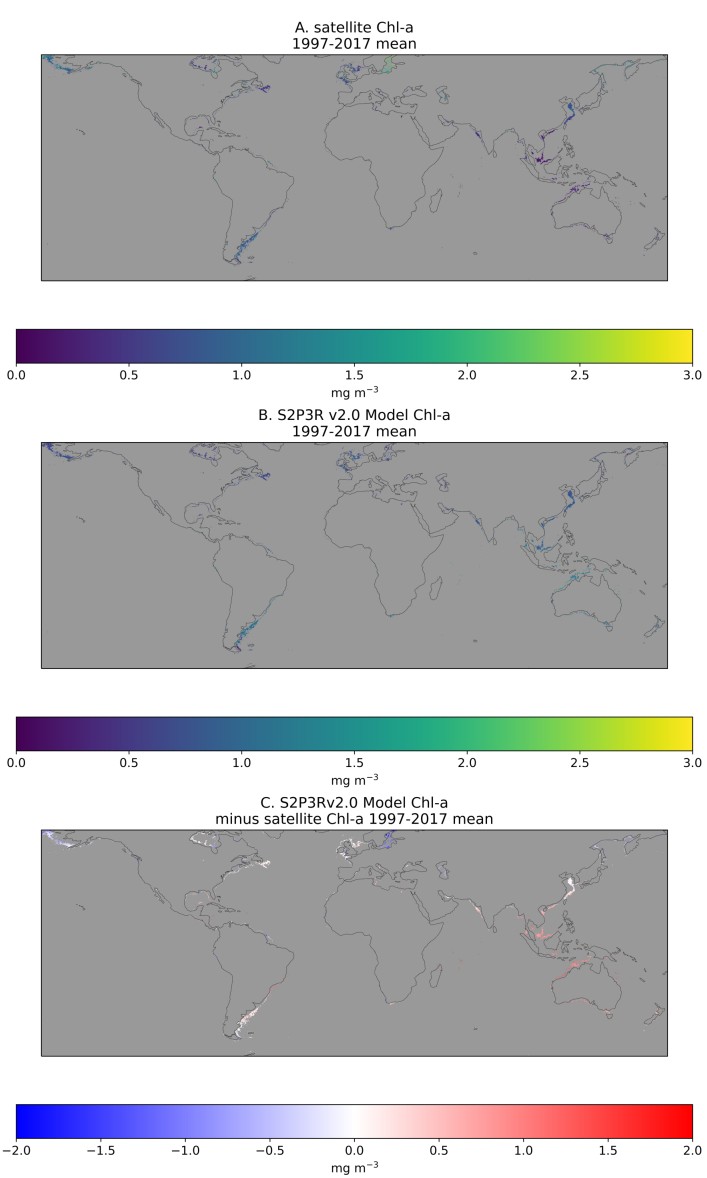





**Figure 9. Comparison of surface level chlorophyll-a concentrations with satellite based chlorophyll-a estimates (Sathyendranath *et al.*, 2020). Figures present an annual mean of all data available between 1997 and 2017 inclusive. Satellite data filtered to include just case 2 water, i.e. water ≥ 70m water depth (Jackson et al., 2019). The nutrient data to which the water in the model's bottom level was relaxed to is taken from the winter values in World Ocean Atlas for each hemisphere.**

To facilitate a more detailed understanding of the model performance, we now evaluate the model in one mid-latitude region, the North West European Shelf, then one lower latitude region, the Great Barrier Reef.

**6.1 North West European Shelf Physical Evaluation**

The North West European Shelf is both typical of the mid-latitude regions, where the assumptions made in this modelling framework appear to work well (Figure 4), and is a large area of shallow water which has previously been studied in detail both observationally (e.g. Smyth *et al.*, 2015) and using state of the art 3D models (e.g. Graham *et al.*, 2018).

Forced with the ERA5 atmospheric data (Hersbach *et al.*, 2019), S2P3-R v2.0 simulates the time-averaged SST within 0.5K across much of the North West European Shelf (Figure 10). The model also simulates the trend and interannual variability in SST well in the North Sea, English Channel and Irish Sea (Figure 11), despite the North Sea and English Channel displaying cool and warm temperature biases of approximately 0.5K respectively (Figure 11). The cool bias in the northern North Sea is consistent with the model not accounting for the inflow of relatively warm Atlantic Water via the Dooley Current between Orkney and Shetland (Dooley, 1974; Marsh *et al.*, 2017; Sheehan *et al.*, 2020).





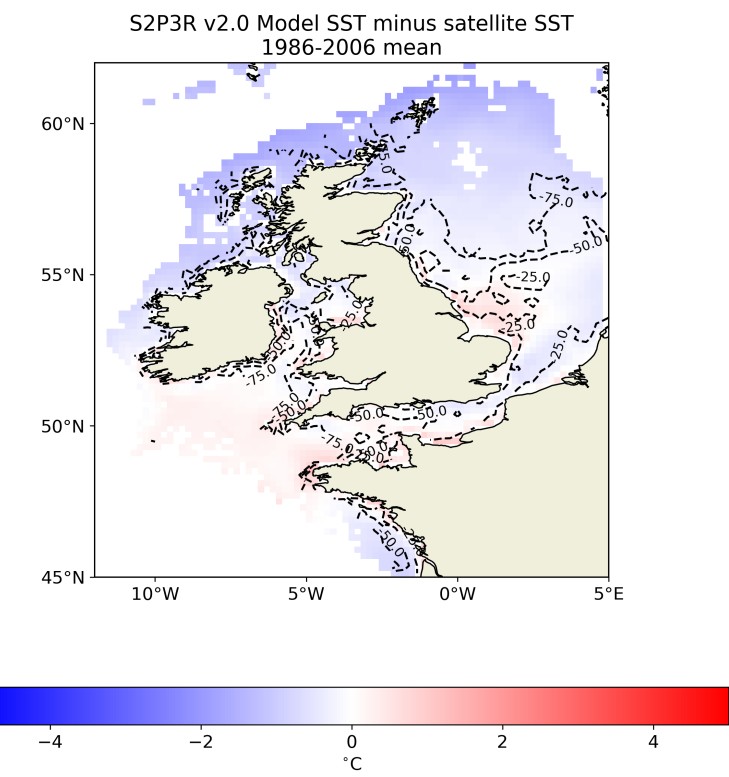

2   **Figure 10. S2P3R v2.0 SST averaged between the years 1986 and 2006 inclusive minus satellite SSTs (Merchant et al.,**

3   **2019) averaged over the same interval. Labelled dashed lines illustrate bathymetry in meters.**

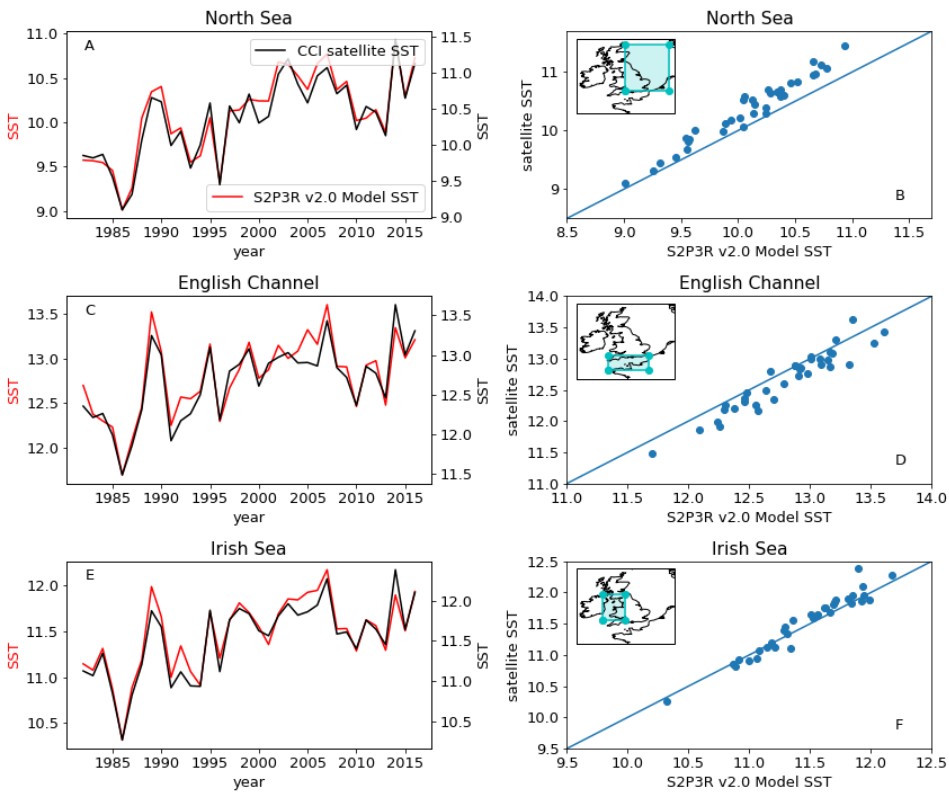

**Figure 11. S2P3R v2.0 SST averaged annually and across the three regions highlighted in inset maps, and annually**

**averaged satellite SSTs (Merchant et al., 2019) from the same regions.**

Bottom water temperatures can be examined at individual locations using mooring data, as done in Marsh et al., (2015), or at sparse locations against gridded data (e.g. Good, Martin and Rayner, 2013), but to facilitate a more spatially complete assessment we here turn to state-of-the-art model output, generated by the 1.5km NEMO-shelf Atlantic Margin Model (AMM15) (Graham *et al.*, 2018). We find that S2P3-R v2.0 replicates the average values and interannual variability in bottom water temperatures in the North Sea, English Channel and Irish sea captured by the AMM15 model (Graham *et al.*, 2018) with biases of less than 0.5K and $R^2$ values of 0.92, 0.84 and 0.93 the North Sea, English Channel and Irish Sea respectively (Figure 12). While the AMM15 model is not a perfect surrogate for observations, this comparison gives us confidence in these regions that the use of the highly computationally efficient S2P3-R v2.0 model to a first order gives us comparable bottom water temperature results to a state-of-the-art and computationally demanding three-dimensional modelling system.

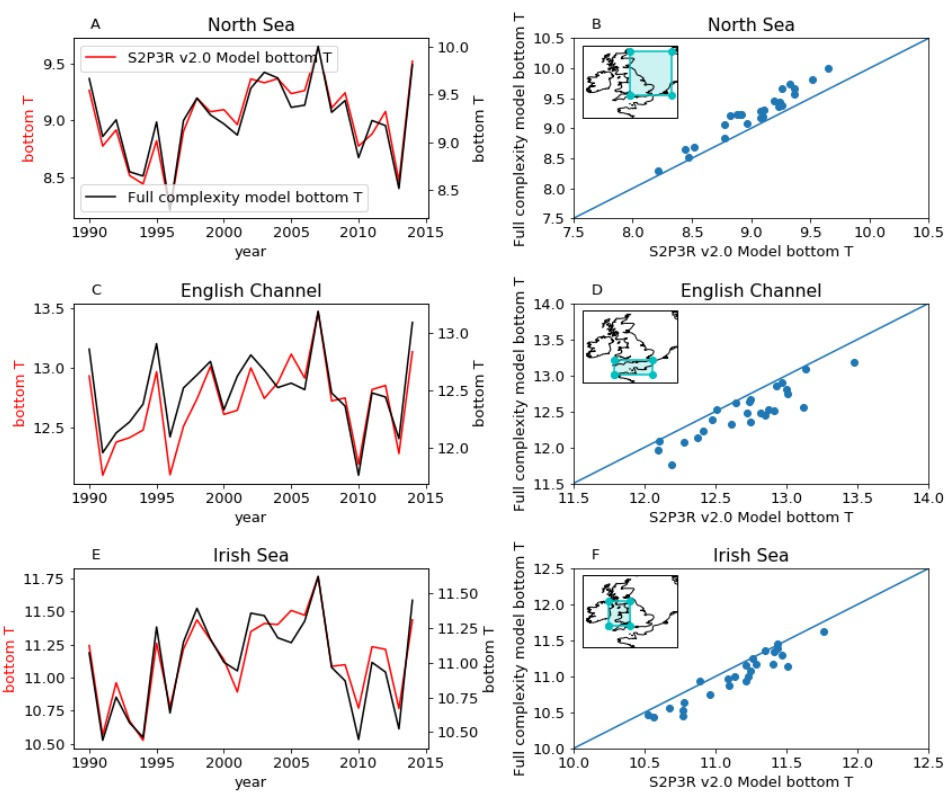

**Figure 12. S2P3R v2.0 bottom water temperatures averaged annually and across the three regions highlighted in inset maps, and annually bottom water temperatures from these same regions taken from a state-of-the art shelf sea model hindcast (Graham et al., 2018).**

**6.2 North West European Shelf Biogeochemical Evaluation**





1 S2P3R v2.0 underestimates surface chlorophyll-a when compared to annual mean satellite derived estimates

2 (Sathyendranath *et al.*, 2020) across most of the North West European Shelf by 0.25 to 0.50 mg/m$^3$ (Figure 13).

3 The smallest bias is seen in the North Sea, and the largest in the Irish Sea (Figure 13).

**Figure 13. Comparison of North West European Shelf surface level chlorophyll-a concentrations with satellite based chlorophyll-a estimates (Sathyendranath et al., 2020). Figures present an annual mean of all data available between 1997 and 2017 inclusive. Dashed lines represent 20m depth contours. Satellite data are filtered to include just case 2 water, i.e. water ≥ 70m water depth (Jackson et al., 2019).**

The seasonal and interannual variability of phytoplankton production, and therefore chlorophyll-a concentration
are strongly influenced by changes in stratification. Where the water column is mixed throughout the year (e.g
English channel and southern North Sea), phytoplankton growth tends to display a single peak governed to a first
order by the cycle of solar irradiance and the availability of nutrients, with development of the peak slowed by
mixing of phytoplankton into deeper, poorly lit, waters (e.g. Figure 14 a, c, e, g, i, j) (Wafar, Le Corre and Birrien,
1983). Where the water column is seasonally stratified and winter mixing has removed any upper-water column
nutrient limitation potential, a spring bloom typically develops as the mixed layer - defined by turbulence levels
(Chiswell, 2011; Chiswell, Calil and Boyd, 2015) - shallows across a seasonally-deepening critical depth,
shallower than which light-limited phytoplankton production exceeds approximately depth-invariant
phytoplankton losses (Sverdrup, 1953). In these seasonally stratified waters, an autumn bloom (and therefore
second chlorophyll-a peak) may also develop as cooling results in buoyancy loss from the surface or winds
increase turbulence, and the mixed layer deepens and refreshes what have become nutrient limited sunlit waters,
with nutrients from deeper in the water column (Findlay *et al.*, 2006). This potentially skewed, bimodal
distribution is captured by the model in seasonally stratified sites (Figure 14, c, k, o). While in the central North
Sea and Celtic Sea, the seasonal evolution of model chlorophyll-a concentrations match closely with that inferred
from observations (Figure 14 c, k, o), in most sites the model fails to capture the full complexity of the seasonal
signal. The model also fails to capture the interannual variability in chlorophyll-a at those sites where long enough
observational timeseries exist to assess this (Figure 14). The lack of evidence for correctly simulated interannual
variability potentially reflects the importance of processes not represented in this model such as photo-acclimation
(Bahamondes Dominguez *et al.*, 2020), grazing (Bahamondes Dominguez *et al.*, 2020) and phytoplankton species
composition (Barnes *et al.*, 2015) in controlling interannual variability, or the importance of variability in nutrient
flux across the shelf-break (Holt *et al.*, 2012) and from rivers (Capuzzo *et al.*, 2018).





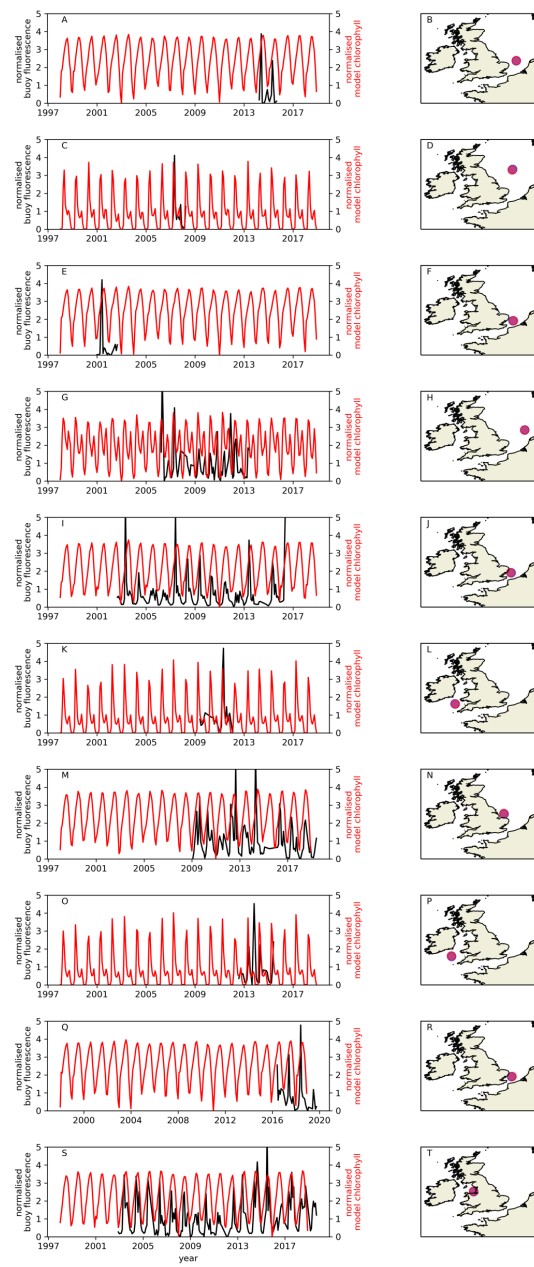

**Figure 14 Comparison of model chlorophyll-a timeseries (red) with chlorophyll-a fluoresence measurements (black)**

**made on 10 autonomous buoys situated round the UK as part of the CEFAS SmartBuoy network (Sivyer, 2016). Both**

**datasets have been monthly averaged and normalised by their standard deviation. Fluoresence data has been filtered**





**to include only that collected between 18:00 and 06.00 hours to avoid quenching of the signal by sunlight. Maps on the**
**right hand side illustrate the location of each buoy. Note that buoys have been operational over different time windows.**
**7.1 Great Barrier Reef Physical Evaluation**
Moving to the low-latitudes where SST biases in the S2P3R v2.0 model are typically larger than they are in the
mid-latitudes (Figure 4), a simulation has been undertaken which encompasses the Great Barrier Reef (GBR).
The GBR is well instrumented, allowing analysis of subsurface as well as surface temperatures in this region.
The modelled SSTs in the GBR display a positive bias relative to satellite SSTs in the north and negative bias in
the south (Figure 15). This may relate to the fact that the model does not simulate lateral advection, which will be
exporting heat from the north to the south in the East Australian Current.
S2P3R v2.0 appears to capture much of the interannual variability observed in SSTs over the GBR since the early
1980s (Figure 16), but with a temperature bias of <0.5K (Figure 15). The simulation however appears to
erroneously simulate a stepwise cooling around the year 2000 which compromises the overall correlation between
model and satellite SSTs (Figure 16). This stepwise cooling may reflect changes in the assimilation of
observations into the ERA5 reanalysis product which is used to force the model.



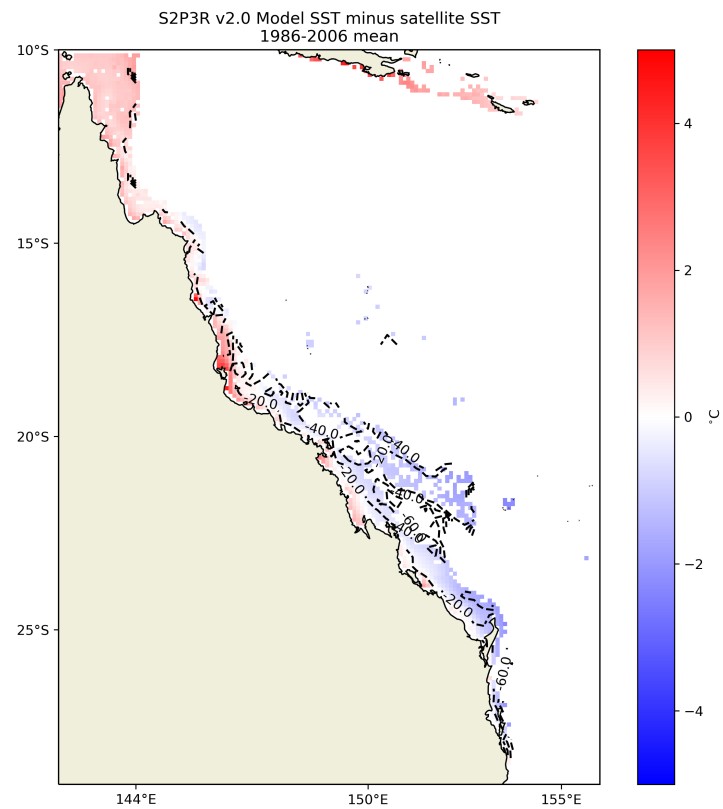

2 **Figure 15. S2P3R v2.0 SST averaged between 1986 and 2006 inclusive minus satellite SSTs (Merchant et al., 2019) over**

3 **the same interval. Labelled dashed lines show bathymetry in meters.**

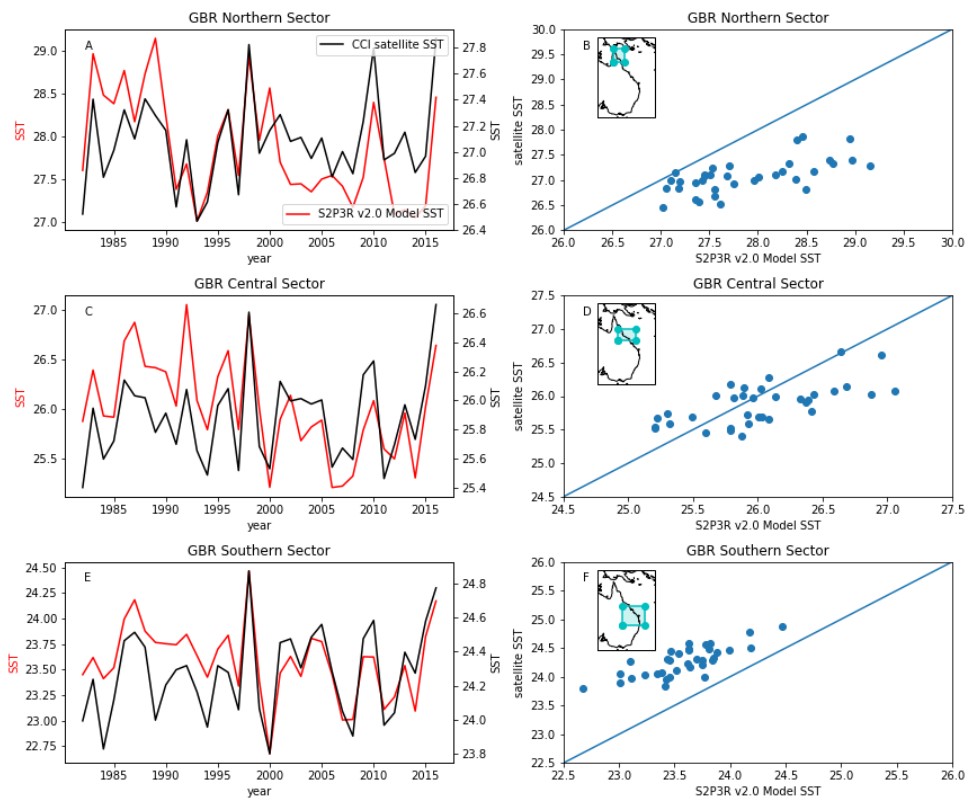

**Figure 16. Comparison of interannual SST variability between S2P3R v2.0 and satellite (Merchant et al., 2019) over the GBR, subdivided into three latitudinally delineated regions. These regions are identified in the inset maps.**

While a state-of-the-art regional model for the GBR region exists (Steven et al., 2019), a long validated hindcast is not available to allow evaluation of the S2P3R V2.0 simulation of bottom water temperatures in the GBR analogous to that presented here for the North West European Shelf. The GBR is however instrumented with an extensive mooring network, making up part of the IMOS FAIMMS (Integrated Marine Observing System, Facility for Automated Intelligent Monitoring of Marine Systems) Sensor Network (IMOS, 2009a, 2009b, 2009d, 2009e, 2009c, 2015, 2017). The location of the moorings utilised in the evaluation presented here are highlighted in Figure 17.

In situ observations indicate that a cool bias exists in the modelled SSTs, but this is restricted to austral winter months (Figure 17c). The fact that a warm bias is not evident in the mooring data, as it is in the satellite SST data (Figure 15), may result from a sampling bias within the mooring dataset towards deeper waters. Modelled bottom water temperatures from the lower latitude mooring sites present a cool bias, but a linear relationship when compared with observational data (Figure 17d). The cool bias may reflect the fact that the model output against which the observations are compared represent a mean value across a ~10km$^2$ grid-cell and for example may well therefore not be simulating the conditions at the same depth as the observations are made.



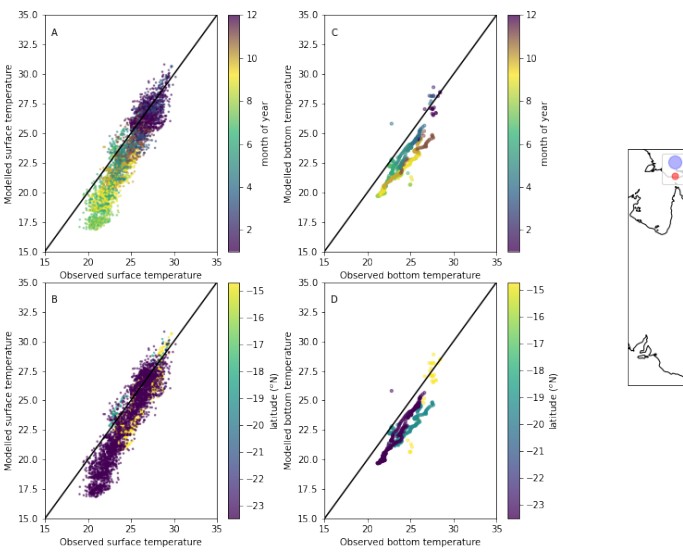

**Figure 17. Comparison of S2P3R v2.0 surface (A and B) and bottom (C and D) temperatures against mooring observations from IMOS and FAIMMS and moorings (IMOS, 2009a, 2009b, 2009d, 2009e, 2009c, 2015, 2017). The x-y values of the data presented in plots on the top and bottom (A and B, and C and D) are identical, but are coloured to highlight where temporal and geographical biases exist in the model output. Seabed observations are considered here to be those falling within 5m of the site-depth for each mooring. The map shows the location of surface (red) and bottom (blue) temperature mooring observations used in model evaluation over the GBR.**

**7.2 Great Barrier Reef Biogeochemical Evaluation**

In GBR case 2 water, i.e. water ≥ 70m water depth (Jackson et al., 2019), where comparison with the European Space Agency Climate Change Initiative (ESA CCI) long term satellite chlorophyll data (Sathyendranath et al., 2020) is meaningful, the S2P3R v2.0 simulation of chlorophyll displays low negative biases <0.2 mg/m$^3$ (Figure 18). These biases are considerably lower than those simulated on the North West European shelf (Figure 13), the region within which the model was originally designed to investigate chlorophyll seasonality (Sharples *et al.*, 2006).

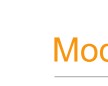
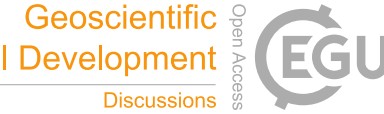

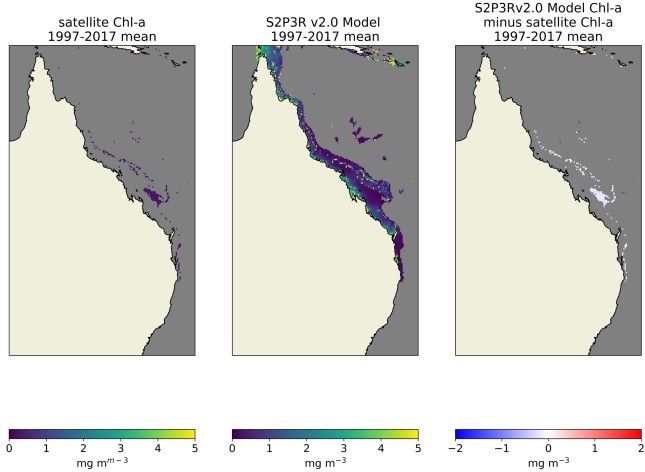

**Figure 18. Comparison of GBR surface level chlorophyll-a concentrations with satellite based chlorophyll-a estimates**

**(Sathyendranath *et al.*, 2020). Figures present an annual mean of all data available between 1997 and 2017 inclusive.**

**Satellite data filtered to include just case 2 water, i.e. water ≥ 70m water depth (Jackson et al., 2019).**

Evaluation of the model's ability to simulate the seasonal cycle and interannual variability in chlorophyll-a in the GBR region has been conducted using moored buoy fluorescence data, as done for the North West European Shelf, but with more restricted temporal coverage (Figure 19). Unlike the spring/autumn bloom dominated seasonal evolution of chlorophyll-a experienced in many temperate sites, the seasonal cycle simulated by the model and illustrated by the observations across the GBR sites examined here follows a relatively smooth oscillation with the peak values in the model data occurring in late summer (Figure 19). In contrast to many of the North West European Shelf sites, this behaviour likely results from the intersection of the critical depth (Sverdrup, 1953) with the seabed at these high light and shallow locations. The incomplete or short length of the GBR fluorescence observational datasets mean that it is not possible to undertake a detailed investigation of interannual variability, however the longest of the mooring datasets (Figure 19k) exhibits its lowest chlorophyll-a peaks in the same years as those simulated by the model (2018 and 2019). In a typically oligotrophic setting, like much of the GBR, one might expect year to year variability to be dominated by injections of nutrients from the shelf-break or the coast (Furnas and Mitchell, 1986). Despite the model not representing these processes, it never-the-less simulates considerable inter-annual variability, indicating the potential for atmospheric and vertical ocean dynamics drivers of such variability.

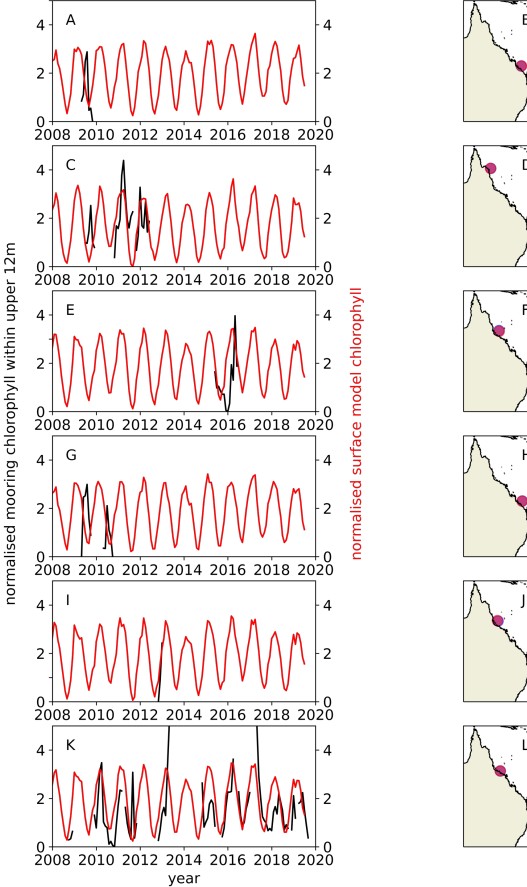

**Figure 19. Comparison of model chlotophyll-a timeseries (red) with chlorophyll-a fluoresence measurements (black) made on 6 moored buoys situated down the GBR as part of the IMOS Australian National Mooring Network (IMOS, 2015). Both datastes have been monthly averaged and normalised by their standard deviation. Surface level data was not available for all sites, so data represent an average over the top 12m of the water column to improve spatial data coverage.**

## 8. Summary and discussion:

Forced by observation-derived atmospheric conditions, a simulation spanning the shelf seas of the global tropical-to-subpolar ocean at approximately 10km$^2$ resolution captures >50% of the observed interannual SST variability between 2006 and 2016 in ~60% of the grid cells, and greater than 80% of the interannual SST variability in ~20% of the grid cells (Figure 8). This tells us that a large part of the SST variability in a significant component of our global shelf-seas is atmospherically forced, rather than being forced by variable lateral exchanges with the deep ocean or runoff. When compared to satellite data (Merchant *et al.*, 2019), 61% of grid cells however present an SST bias of greater than 1K and 42% present an SST bias of greater than 2K, highlighting limitations to the simple modelling approach.




Together, analysis of SST variability and SST bias indicate there are significant areas of our global shelf seas
where the model should be used with extreme caution. These regions are likely to be those which have; (1)
substantial exchange of heat with the open ocean through lateral advection, (2) low tidally driven mixing and
therefore a low ratio of vertical/horizontal control over SSTs (Figure 5), (3) significant influences from local
processes/properties such as riverine inputs or locally unusual bottom drag coefficients, (4) high salinity
variability and low temperatures, or (5) on-shelf propagation and dissipation of the internal tide. The model could
however be tuned to account for some of these influences if studies were to be undertaken with a focus on such
regions.
Regional evaluation has been conducted across the North West European Shelf around the UK, and the Great
Barrier Reef. The model captures most of the observed SST trend and variability in the waters around the UK
(Figure 11), with a temperature biases of <0.5K across most of the region (Figure 10). S2P3R v2.0 also captures
between 84 and 93% of the variability in bottom water temperatures simulated by a state-of-the-art shelf sea model
hindcast (Graham *et al.*, 2018) for the three focal regions of the North Sea, English Channel and Irish Sea.
Comparison of modelled and satellite SSTs across the Great Barrier Reef indicate over ~10-year intervals the
model performs well, but there appear to be step-changes in the modelled SST which are not seen in the satellite
data. The discontinuity occurring around the year 2000 may reflect a step change in the data assimilation
configuration used within the ERA5 product or data being assimilated by that product (Hersbach *et al.*, 2018)
used to provide the atmospheric forcing to S2P3R v2.0. Alternatively, the step changes may result from changes
in the lateral supply of heat from the open ocean.
A particular strength of this modelling approach is likely to be in examining or predicting anomalies or extremes
which occur under a consistent set of oceanographic conditions. For example, the marine heat waves associated
with tropical coral bleaching tend to occur following doldrum-like condition, when there is limited advection and
mixing (Skirving, Heron and Heron, 2011).
Observational limitations mean the model's simulation of biological production in space and time is harder to
assess than that of temperature. The model however captures the broad scale patterns of surface chlorophyll
(Figure 9, Figure 13, Figure 18), with a weak indication of latitudinally varying bias towards overprediction in
low latitudes and underprediction in high latitudes (Figure 9). The model displays considerable skill in many
locations at simulating intra-annual chlorophyll variability (Figure 14, Figure 19), but no demonstrable skill at
simulating inter-annual chlorophyll variability. This implies that the large-scale processes which govern the
seasonal progression of primary production do not also govern interannual variability. Factors such as riverine
input of nutrients may dominate interannual variability in many locations (Lenhart, Radach and Ruardij, 1997).
These results emphasise the importance of decadal and longer observational biogeochemical timeseries for
assessing the skill of models at simulating those processes which are likely to govern the biogeochemical response
of our shelf seas to anthropogenic climate change.
In summary, S2P3R v2.0 is a simple to use, computational efficient shelf sea modelling tool ideally suited to (a)
semi-dynamically downscaling climate projections, (b) undertaking large-scale, long or large ensemble
projections, (c) use by management or policy groups without access to large technical or computational resources.





The objective assessment of the model presented here will hopefully guide potential users as to whether S2P3R
v2.0 is the tool to answer their questions. Where S2P3R v2.0 is considered to be an appropriate tool, we would
encourage local assessment of the data presented here at a global scale and hope to facilitate this through the
provision of these data (see Data Availability section). Finally, within the Code Availability section of this
manuscript we provide the model code, code required to produce the model forcing datasets, an example model
setup with pre-prepared forcing data, and within the Readme file, step-by-step instructions for setting up and
running the model.
**9. Code Availability**
S2P3Rv2.0 is available from github: https://github.com/PaulHalloran/S2P3Rv2.0
The release associated with this manuscript (https://github.com/PaulHalloran/S2P3Rv2.0/releases/tag/v1.0.1) has
been archived to Zenodo with the following DOI 10.5281/zenodo.4147559.
The README file available on github or via the DOI link provides step by step instructions for how to install,
setup and run the model, and provides a basic script for analysing the model output. At the bottom of the README
a worked example is provided to help the user go through the full process from generating model forcing files,
running the model and displaying the output with some example data.
**10. Data availability**
Model minus satellite SST data from the global (65ºS-65ºN) simulation averaged between 2006 and 2016 from
which the global validation has been undertaken in this manuscript are archived as netCDF and CSV files to allow
potential users to undertake bespoke assessment of the model http://doi.org/10.5281/zenodo.4018815 (Halloran,
20  2020).

**11. Competing interests**
The authors declare that they have no conflict of interest.
**12. Author contribution**
The model development was undertaken by PH. PH lead the analysis with contributions from JMW, BN, RM
and WS. All authors contributed to the writing of the manuscript.
**13. Acknowledgements**

Paul Halloran was supported by the UK Research Council grant NE/V00865X/1. Paul Halloran and Jennifer
McWhorter were supported by the QUEX Institute, a University of Exeter and University of
Queensland Partnership. This project has received funding from the European Union's Horizon 2020 research and
innovation programme under grant agreement No 820989 (project COMFORT, Our common future ocean in the
Earth system – quantifying coupled cycles of carbon, oxygen, and nutrients for determining and achieving safe



operating spaces with respect to tipping points). This manuscript reflects only the authors' views; the European
Commission and their executive agency are not responsible for any use that may be made of the information the
work contains. William Skirving was supported by NOAA grant NA19NES4320002 (Cooperative Institute for
Satellite Earth System Studies) at the University of Maryland/ESSIC. The scientific results and conclusions, as
well as any views or opinions expressed herein, are those of the author(s) and do not necessarily reflect the views
of NOAA or the Department of Commerce. The Smartbuoy data was made available by Cefas and funded
by Defra and the UK Research Council Candyfloss and Celtic Deep 2 grant NE/K001957/1. The IMOS
buoy data was provided by the IMOS Queensland and Northern Australian Moorings sub-facility of the
Australian National Mooring Network funded by the Australian Institute of Marine Science and the Integrated
Marine Observing System (IMOS)—IMOS is enabled by the National Collaborative Research Infrastructure
Strategy (NCRIS), supported by the Australian Government. It is operated by a consortium of institutions as an
unincorporated joint venture, with the University of Tasmania as Lead Agent. www.imos.org.au.

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
