# Peer review of "S2P3-R v2.0: computationally efficient modelling of shelf seas 1"

_Geoscientific Model Development, 2021_

## Author Comment (AC1)

**Response to Anonymous Referee #1 comments on gmd-2021-2**
**"S2P3-R v2.0: computationally efficient modelling of shelf seas on regional to global scales"**

Anonymous Referee #1

Referee comments are replicated in full in *back italics*.

Author responses to comments are in *blue italics*.

**General comments**

*The overall quality of the preprint is good, and the described developments of the S2P3-R model are suitable for publication in this journal. The manuscript describes novel model updates and evaluations of the model both globally and regionally. The information and data provided will allow others to assess if the model may be appropriate for their use.*

*We thank the reviewer for taking the time to undertake this review and for their constructive comments. We are pleased to see the reviewer supports publication of the manuscript in GMD.*

**Specific comments**

*No comparison is made of regional biogeochemical performance compared to a long reanalysis such as the Copernicus Marine Service NORTHWESTSHELF_REANALYSIS_BIO_004_011 product. This seems like a missed opportunity given the spatial coverage compared to satellite data.*

*Thank you for this suggestion, we will carefully look into this product and consider how best to make use of it.*

**Suggested minor revisions**

*Comparisons to satellite data state data was limited to "case 2 water, i.e. water ≥ 70m water depth (Jackson et al., 2019)". Whether this is meant to be "case 1 water" or "≤ 70" isn't clear as the reference "Jackson et al 2019" does not seem to be available. I would suggest a clarification of satellite data selection criteria.*

*Thank you for highlighting the mistake in the manuscript. This should read "case 1 waters". This will be updated in the revised manuscript.*

*We are sorry that that you were unable to accesss the Jackson et al., 2019 reference. This is a user guide for the satellite product and as such unfortunately does not have a DOI. The suggested link takes you to the repository which now contains more recent version of the documentation. We will resolve this as best we can in the bibliography of the revised manuscript, seeking guidance from the journal about how best to reference such a document.*

The data variability in Figure 9A and 9B (and to a lesser extent Figure 18) is difficult to distinguish with a grey background. I would suggest the use of a white background for plots with viridis colourmap, such as in Figure 7. These global plots would also benefit from being larger, single viridis and blue-white-red colour bars could be positioned either side.

*Thank you for this suggestion. The grey background was employed to emphasise the data when using a blue-white-red colour palette, but you are absolutely right, it is not good to use it where we have employed the viridis colour palette.*

*We will change the background colour for plots using the viridis colour palette and make the global plots as large as possible.*

Figure 13 would benefit from enlargement and using a log scale may be more appropriate.

*We agree that a log scale makes more sense for the comparison. We will do this, thank you for the suggestion. We will also make this figure as large as possible.*

Technical corrections

Page 12 Line 21: suggest replacing "other" with "apart from"

*This change will be made.*

Page 29 line 41: This reference doesn't appear to be available from the url provided.

*We are sorry that the link to the Jackson et al., 2019 reference did not work. As mentioned above, this is a user guide for the satellite product and as such unfortunately does not have a DOI. The suggested link takes you to the repository which now contains more recent version. We will resolve this as best we can in the bibliography of the revised manuscript.*

**Response to Anonymous Referee #2 comments on gmd-2021-2**

*"S2P3-R v2.0: computationally efficient modelling of shelf seas on regional to global scales"*

Anonymous Referee #2

**General comments**

The paper presents an upgrade to a previously published modelling system. Addressing forcing issues that allow the system to be run over larger areas and longer times. I believe this to be a sufficiently large advance in modelling science to merit publication. The methods are clear and well presented. The protocols appear to be well documented with the supplied code (though I have not tested them).

The document presents an honest accounting for the strengths and weaknesses of the modelling system. In places this is a little too sweeping, or lacking the detail that would permit the reader to make scientific inferences from the results. E.g. to what extent does the exhibited skill over the Patagonian or North West European Shelf imply that later fluxes are not important. But that is not the aim of the paper.

The results support the concluding remarks, except that I would more strongly state the possible value of this tool in education (perhaps to undergraduates?). I also think that the value to policy groups of this "cheap" model is perhaps slightly dangerous if the output are not in some way corroborated with existing data from higher-expense simulations. After all this paper, at length, highlights the gains in efficiency do come with a loss in skill.

*We thank the reviewer for their time undertaking the review and providing these valuable comments, and are pleased to see that they support publication of this work.*

*We will broaden the discussion within the revised manuscript to encompass the two excellent points around the use of the tool in education, and caveating the suggestion that the tool could be valuable for policy groups.*

**Specific comments**

p9. Fig 3 caption: line 6: Without parenthetic commas, the "therefore" comes in the wrong place. E.g. This might be clearer:

"Where this is positive there is a net heat flux into the ocean. So, assuming the system is approximately at steady state, advection of heat is therefore out of this area."

*Thank you for the suggestion, this change will be made.*

p9 line 15: should read "… more prevalent at low M2 tidal amplitudes…"

M2 is dominant in the North West European Shelf, in most places. But K1 can be relatively large in other regions, like the South China Sea

*This is an important clarification, thank you. The change will be made.*

p10. Line 2. Unpack this line. Is it the case? Does Figure 5 exhibit smaller model biases in the summer? Confirm what you think my eyes are telling me.

*Thank you for questioning this. On reflection we should have been explicit about what we mean by mid-latitudes. We are considering these to be the regions 30-60 degrees N/S and will specify this in the revised manuscript. Because the eye can easily be drawn to the tropical areas highlighted in blue off northern Australia and Indonesia, we will explicitly mention the key mid-latitude regions which do show summer biases which are smaller than the winter biases (Scotian Shelf), and contrast this with the majority of regions which show little seasonality in the bias, e.g. NW European Shelf and Patagonian Shelf, and areas which appear to show a stronger summer than winter bias, e.g. South China Sea and Bering Sea.*

---

## Author Response (AR1)

**Response to Anonymous Referee #1 comments on gmd-2021-2 "S2P3-R v2.0: computationally efficient modelling of shelf seas on regional to global scales"**

Referee comments are replicated in full in back italics.

Author responses to comments are in *blue italics*.

A full copy of the revised manuscript is provided at the end of this document with revised text highlighted in red, and revised figures outlined with a red box.

Anonymous Referee #1

General comments

The overall quality of the preprint is good, and the described developments of the S2P3-R model are suitable for publication in this journal. The manuscript describes novel model updates and evaluations of the model both globally and regionally. The information and data provided will allow others to assess if the model may be appropriate for their use.

We thank the reviewer for taking the time to undertake this review and for their constructive comments. We are pleased to see the reviewer supports publication of the manuscript in *GMD*.

Specific comments

No comparison is made of regional biogeochemical performance compared to a long reanalysis such as the Copernicus Marine Service NORTHWESTSHELF\_REANALYSIS\_BIO\_004\_011 product. This seems like a missed opportunity given the spatial coverage compared to satellite data.

We thank the reviewer for the suggestion that we take a look at the North West Shelf biogeochemical reanalysis available through the Copernicus Marine Service. The suggested product is a NEMO-ERSEM reanalysis which assimilates Chlorophyll from the ESA CCI Ocean colour product already used in this manuscript. The product that the reviewer recommends is state-of-the-art and very exciting, but it is the author's opinion that ocean biogeochemical reanalysis products have not yet reached a level of maturity where we can use them as surrogates for observations in the same was as we are beginning to for physical atmospheric and ocean variables. This opinion is supported by limited agreement between the NEMO-ERSEM reanalysis data and in-situ chlorophyll observations, see figure 6 in (Ciavatta et al., 2018). We have therefore stuck to evaluating the North West European Shelf biogeochemistry against satellite and in-situ observations.

**Suggested minor revisions**

Comparisons to satellite data state data was limited to "case 2 water, i.e. water  $\geq$  70m water depth (Jackson et al., 2019)". Whether this is meant to be "case 1 water" or " $\leq$  70" isn't clear as the reference "Jackson et al 2019" does not seem to be available. I would suggest a clarification of satellite data selection criteria.

Thank you for highlighting the mistake in the manuscript. This should have read "case 1 waters".

We are sorry that that the reviewer was unable to access the Jackson et al., 2019 reference. The link was to a user guide for the satellite product and as such unfortunately did not have a DOI. In the revised manuscript we have moved away from this reference to a peer reviewed article (Sathyendranath et al., 2019), and clarified the water depth selection criteria as follows:

"Comparison is made between the S2P3R v2.0 simulation of chlorophyll and annually averaged European Space Agency Climate Change Initiative (ESA CCI) long term satellite chlorophyll data (Sathyendranath et al., 2020). The ESA CCI long term satellite chlorophyll product is focused on case-1 waters (Sathyendranath et al., 2019). The comparison presented here is therefore restricted to water depths  $\geq$  70m a compromise which allows us to exclude the most coastally influenced waters while maintaining moderate spatial coverage."

The data variability in Figure 9A and 9B (and to a lesser extent Figure 18) is difficult to distinguish with a grey background. I would suggest the use of a white background for plots with viridis colourmap, such as in Figure 7. These global plots would also benefit from being larger, single viridis and blue-white-red colour bars could be positioned either side.

Thank you for this suggestion. The grey background was employed to emphasise the data when using a blue-white-red colour palette, but you are absolutely right, it is not good to use it where we have employed the viridis colour palette. Experimenting with different colour backgrounds and different perceptually uniform sequential colour palettes we have settled on a very light grey background and the viridis colour palette. A completely white background presented too little contrast with the yellow data points. The same approach has been applied to figure 18.

Figure 13 would benefit from enlargement and using a log scale may be more appropriate.

We agree that the use of a log scale makes the comparison more straightforward and have changed this. We have also moved the panels close together to allow the figure to be enlarged.

*Extending this suggestion, we have taken both of these points on board to improve figures 14, 18 and 19 as well.*

Specifically, to improve the readability of figure 14 we have split it across two columns, allowing the height of the figure to be increased. We have also narrowed the x-axis range to display just the years where both model data and observational data are present. Figure 19 has been treated in the same way for consistency. Note that we have also modified the normalisation so that instead of subtracting the minimum value then dividing by the standard deviation, we are subtracting the median then dividing by the standard deviation. This approach minimises the impact of extreme points.

**Technical corrections**

Page 12 Line 21: suggest replacing "other" with "apart from"

This typo has been amended, thank you.

Page 29 line 41: This reference doesn't appear to be available from the url provided.

We are sorry that the link to the Jackson et al., 2019 reference did not work. As mentioned above, this is a user guide for the satellite product and as such unfortunately does not have a DOI. We have revised the point we make to allow us to use the published article Sathyendranath et al., 2019, copied below.

Sathyendranath, S. et al. (2019) 'An ocean-colour time series for use in climate studies: The experience of the ocean-colour climate change initiative (OC-CCI)', Sensors (Switzerland). doi: 10.3390/s19194285.

**Response to Anonymous Referee #2 comments on gmd-2021-2 "S2P3-R v2.0: computationally efficient modelling of shelf seas on regional to global scales"**

**Anonymous Referee #2**

**General comments**

The paper presents an upgrade to a previously published modelling system. Addressing forcing issues that allow the system to be run over larger areas and longer times. I believe this to be a sufficiently large advance in modelling science to merit publication. The methods are clear and well presented. The protocols appear to be well documented with the supplied code (though I have not tested them).

The document presents an honest accounting for the strengths and weaknesses of the modelling system. In places this is a little too sweeping, or lacking the detail that would permit the reader to make scientific inferences from the results. E.g. to what extent does the exhibited skill over the Patagonian or North West European Shelf imply that later fluxes are not important. But that is not the aim of the paper.

The results support the concluding remarks, except that I would more strongly state the possible value of this tool in education (perhaps to undergraduates?). I also think that the value to policy groups of this "cheap" model is perhaps slightly dangerous if the output are not in some way corroborated with existing data from higher-expense simulations. After all this paper, at length, highlights the gains in efficiency do come with a loss in skill.

**We thank the reviewer for their time spent undertaking the review and providing these valuable comments. We are pleased to see that they support publication of this work.**

We have broadened the justification text within the revised manuscript to encompass these two excellent points. The text that originally read "The accessibility of S2P3-R v2.0 places it within reach of an array of coastal managers and policy makers" Has been extended to read "The accessibility of S2P3-R v2.0 places it within reach of an array of coastal managers and policy makers, allowing it to be run routinely once set up and evaluated for a region under expert guidance. The computational efficiency and relative scientific simplicity of the tool make it ideally suited to educational applications."

The mention of use by managers and policy makers in the conclusion text has been caveated with the phrase 'use after careful evaluation'.

**Specific comments**

p9. Fig 3 [4] caption: line 6: Without parenthetic commas, the "therefore" comes in the wrong place. E.g. This might be clearer:

"Where this is positive there is a net heat flux into the ocean. So, assuming the system is approximately at steady state, advection of heat is therefore out of this area."

Thank you for the suggestion, this change has been made, noting that the caption referred to is that of figure 4.

p9 line 15: should read "... more prevalent at low M2 tidal amplitudes..."

M2 is dominant in the North West European Shelf, in most places. But K1 can be relatively large in other regions, like the South China Sea

*This is an important clarification, thank you. The change has be made.*

p10. Line 2. Unpack this line. Is it the case? Does Figure 5 [6] exhibit smaller model biases in the summer? Confirm what you think my eyes are telling me.

Thank you for questioning this. For clarification, the figure referenced in that line is figure 6, so we have assumed that the question refers this figure. On reflection we should have been explicit about what we mean by mid-latitudes. We are considering these to be the regions 30-60 degrees N/S and specify this in the revised manuscript, along with adding latitude labels to the figure. Because the eye can easily be drawn to the tropical areas highlighted in blue off northern Australia and Indonesia, we now explicitly mention the key mid-latitude regions which do show such biases and place these in context. In summary, 
[revised manuscript text omitted]